# Exploring the Genetic and Clinical Landscape of Dedifferentiated Endometrioid Carcinoma

**DOI:** 10.3390/ijms26094137

**Published:** 2025-04-27

**Authors:** Hikaru Haraga, Kentaro Nakayama, Sultana Razia, Masako Ishikawa, Hitomi Yamashita, Kosuke Kanno, Mamiko Nagase, Tomoka Ishibashi, Hiroshi Katagiri, Ryoichi Shimomura, Yoshiro Otsuki, Satoru Nakayama, Satoru Kyo

**Affiliations:** 1Department of Obstetrics and Gynecology, Faculty of Medicine, Shimane University, 89-1, Enya-Cho, Izumo 693-8501, Shimane, Japan; hikarusaito310@gmail.com (H.H.); m-ishi@med.shimane-u.ac.jp (M.I.); memedasudasu1103@gmail.com (H.Y.); kanno39@med.shimane-u.ac.jp (K.K.); 2Department of Obstetrics and Gynecology, Nagoya City University East Medical Center, Nagoya 464-8547, Aichi, Japan; tomoka@med.nagoya-cu.ac.jp; 3Department of Legal Medicine, Faculty of Medicine, Shimane University, 89-1, Enya-Cho, Izumo 693-8501, Shimane, Japan; raeedahmed@yahoo.com; 4Department of Pathology, Faculty of Medicine, Shimane University, 89-1, Enya-Cho, Izumo 693-8501, Shimane, Japan; mami55@med.shimane-u.ac.jp; 5Department of Obstetrics and Gynecology, Masuda Red Cross Hospital, I103-1, Otoyoshi-Cho, Masuda 698-8501, Shimane, Japan; jimipin999@gmail.com; 6Department of Pathology, Masuda Red Cross Hospital, I103-1, Otoyoshi-Cho, Masuda 698-8501, Shimane, Japan; shimomura-r@masuda.jrc.or.jp; 7Department of Pathology, Seirei Hamamatsu General Hospital, 2-12-12, Sumiyoshi, Chuo-ku, Hamamatsu 430-8558, Shizuoka, Japan; otsuki@sis.seirei.or.jp; 8Department of Obstetrics and Gynecology, Seirei Hamamatsu General Hospital, 2-12-12, Sumiyoshi, Chuo-ku, Hamamatsu 430-8558, Shizuoka, Japan; satoru@sis.seirei.or.jp

**Keywords:** dedifferentiated endometrioid carcinoma, endometrial carcinoma, whole-exome sequencing, p53

## Abstract

Dedifferentiated endometrioid carcinoma (DDEC) is rare, has a poor prognosis, and the genes responsible for dedifferentiation remain unclear. This study aimed to clarify the characteristics of DDEC in Japanese patients and develop treatment strategies. Eighteen DDEC cases were included; their clinicopathological features and prognoses were analyzed and compared to those of other histological subtypes. The samples were divided into well-differentiated and undifferentiated components; immunostaining and whole-exome sequencing (n = 3 cases) were conducted. The incidence of DDEC was 2.0% among endometrial cancers. The 5-year progression-free survival and the 5-year overall survival for DDEC was approximately 40% and 30%, respectively. Immunohistochemistry showed that 66.7% of patients were mismatch repair deficient. The rate of p53 mutations was higher than that reported in previous studies, and patients with p53 mutations in the undifferentiated components had a poor prognosis. Whole-exome sequencing revealed different gene mutations and mutation signatures between well-differentiated and undifferentiated components. New genetic mutations in undifferentiated regions were uncommon in all three cases. One case (case 1) exhibited homologous recombination deficiency, whereas the other two showed microsatellite instability-high and hypermutator phenotypes. Genetic analysis suggests that immune checkpoint and poly (ADP-ribose) polymerase inhibitors and drugs targeting the p53 pathway may be effective against DDEC.

## 1. Introduction

Dedifferentiated endometrioid carcinoma (DDEC) was first reported in 2006 [1] and was subsequently included in the 2014 World Health Organization (WHO) international classification [2]. Histopathologically, DDEC is characterized by a clear border between low-grade (grades 1 or 2) endometrioid and undifferentiated carcinoma [3]. The incidence of DDEC accounts for approximately 1–9% of all endometrial cancers [4,5]. The well-differentiated component (WC) is located superficially, whereas the undifferentiated component (UC) is located deeper, thereby making biopsies prone to diagnostic errors [6]. Furthermore, undifferentiated carcinoma can be misdiagnosed as grade 3 endometrioid carcinoma [7]; therefore, the actual frequency of DDEC may be higher than that previously reported.

The development of new treatment strategies for DDEC is urgently needed owing to its poor prognosis [8,9]. Especially, SWI/SNF-deficient DDEC has a poor prognosis [10], and the loss of E-cadherin and fascin expression is associated with tumor aggressiveness [11]. Recent classifications of endometrial cancer, including The Cancer Genome Atlas (TCGA) and the Proactive Molecular Risk Classifier for Endometrial Cancer (ProMisE) are crucial for guiding postoperative adjuvant therapy and predicting prognosis [12,13,14]. DDEC displays genetic characteristics distinct from typical endometrial cancer, which can vary considerably between cases [15,16]. For example, DDEC rarely expresses hormone receptors even though endometrial cancer is generally hormone-dependent. Deficiencies in mismatch repair (MMR) proteins [17,18] and the SWI/SNF complex [19] are frequently observed in DDEC and associated with prognosis and treatment. Therefore, it may be necessary to evaluate the genomic profile of each patient and use targeted therapies. The exome sequencing of undifferentiated and dedifferentiated carcinomas showed that the *PTEN* mutation was most frequent [15], and another report found somatic mutations in *PIK3CA* (50%), *CTNNB1* (30%), *TP53* (30%), *FBXW7* (20%) and *PPP2R1A* (20%) [20].

Although some recent review articles have explored DDEC [21,22], the genetic abnormalities associated with its development and dedifferentiation remain unclear. Furthermore, no studies have specifically focused on Japanese patients. Therefore, this study aimed to characterize DDEC by collecting data from Japanese patients and identifying genetic abnormalities involved in the dedifferentiation process.

## 2. Results

### 2.1. Natural Incidence Histology and Clinicopathological Characteristics

The natural incidences of histology between 2011 and 2020 are shown in Table 1.

The clinicopathological features of DDEC were compared to those of other cancer types (Table 2). No significant differences were observed in age or between DDEC and carcinosarcoma. DDEC exhibited a higher rate of advanced-stage cancers, greater muscle invasion, and a higher rate of lymphovascular space invasion than grade 1/2 endometrioid carcinoma and other carcinomas. Lymphovascular space invasion was not significantly different from grade 3 endometrioid carcinoma. Lymph node metastasis was significantly less frequent in grade 1/2 endometrioid carcinomas than in other cancer types.

### 2.2. Prognostic Analysis Using the Kaplan–Meier Method

The 5-year PFS and OS rates for DDEC were approximately 40% and 30%, respectively, with a significantly poorer prognosis than grade 1/2 endometrioid carcinoma, grade 3 endometrioid carcinoma, and other carcinomas. No significant differences in prognosis were observed between DDEC and carcinosarcomas (Figure 1A,B).

### 2.3. Immunohistochemical Findings

Immunohistochemistry was performed on 18 DDEC samples divided into WCs and UCs. The positive and negative examples for each staining are shown in Figure 2, and the overall staining results are summarized in Figure 3. Epithelial markers such as PAX8, cytokeratin, and epithelial membrane antigen were positive in all WCs but significantly negative in UCs (*p* = 0.020, 0.041, 0.041, respectively). The proportion of p53 mutations tended to be higher in UCs (87.5%) than in WCs (68.6%) (*p* = 0.083). E-cadherin expression was positive in all WCs but absent in UCs (*p* = 0.000). Fascin expression was significantly higher in UCs than in WCs (*p* = 0.041). The proportion of ARID1A-negative patients was 37.5%. MMR deficiency was observed in 66.7% of cases. Prognostic analysis was conducted separately based on the UCs for p53, E-cadherin, fascin, and ARID1A (Appendix A). For example, two cases of wild-type p53 and 14 cases of p53 mutant in UCs were compared. The same comparison was made for E-cadherin, fascin, and ARID1A, according to the UC results. The p53 mutant group considerably had a poor prognosis (*p* = 0.153, 0.178). The prognosis was significantly worse in the E-cadherin-positive group than in the E-cadherin-negative group (*p* = 0.004, 0.030). There was no significant difference in prognosis depending on the expression of fascin and ARID1A.

### 2.4. Whole-Exome Sequencing

Whole-exome sequencing was performed on well-preserved DDEC tissues from three patients. Integrated molecular and clinical characteristics of the three DDEC cases are summarized in Table 3, including whole-exome sequencing results, TCGA molecular classification, and survival outcomes. Based on the results of whole-exome sequencing, TCGA molecular classification was assigned as follows: case 1 was categorized as copy number high due to the presence of a *TP53* mutation; case 2 was classified as microsatellite instability-high (MSI-H) based on mismatch repair deficiency (MMRd); and case 3 was assigned to the POLE ultramutated subgroup due to the detection of a *POLE* mutation. The variant positions and allele frequencies are summarized in Appendix A. In all three cases, the WC and UC displayed distinct gene mutations. New mutations in the UC were uncommon across the three cases. In case 1, *PIK3CA* amplification, the loss of the heterozygosity of *MLH1* and *PTEN*, and the uniparental disomy of *p53* were observed. Cases 2 and 3 exhibited MSI-H and high frequencies of nonsynonymous single-nucleotide variants, suggesting hypermutator phenotypes. The mutation signatures differed between the WC and UC (Figure 4A,B and Appendix A). The copy number plot of case 1 indicates homologous recombination deficiency (HRD). The copy number plots for cases 2 and 3 were normal (Figure 5A,B and Appendix A).

## 3. Discussion

To the best of our knowledge, this is the first work to collect data on DDEC in Japan. In this study, DDEC accounted for 2.0% of all uterine cancers, aligning with the findings of previous reports [4,5]. The clinicopathological characteristics of DDEC include a high rate of advanced-stage cancer, the muscle layer invasion of more than half of the myometrium, and vascular invasion, all of which are risk factors for recurrence and contribute to a poor prognosis.

Undifferentiated and dedifferentiated endometrioid carcinomas (UDECs) have a 5-year PFS rate of 80% for stage I/II, 29% for stage III, and 10% for stage IV and a 5-year OS rate of 84% for stage I/II, 38% for stage III, and 12% for stage IV [9]. DDEC also has a worse prognosis than grade 3 endometrioid carcinomas, with a reported 2-year OS of 31.3% for DDEC compared to 82.8% for grade 3 endometrioid carcinoma [23]. In a study involving 443 patients with UDEC, the median OS was 14 months, with a 5-year OS rate of 44% [24]. However, no specific PFS or OS data for DDEC have been published or compared with other cancer types. In this study, DDEC exhibited the poorest prognosis compared with grade 1/2 endometrioid carcinoma, grade 3 endometrioid carcinoma, other carcinomas, and carcinosarcoma. The 5-year PFS for DDEC was approximately 40%, and the 5-year OS was approximately 30%, both of which were significantly worse than those of other cancer types.

Immunohistochemical analysis revealed distinct protein expression patterns between the WC and UC of DDEC. The loss of epithelial markers in the UC is a hallmark of DDEC [25]. Furthermore, a high proportion (66.7%) of cases were MMR-deficient. The rate of p53 mutations in DDEC was higher than previously reported [26,27], suggesting that DDEC may resemble Type 2 endometrioid carcinoma [28] or p53-abnormal endometrioid carcinoma according to TCGA molecular classification [12]. This finding may explain the poor prognosis observed in the present study. Prognostic analyses of p53, E-cadherin, fascin, and ARID1A revealed no significant differences between fascin and ARID1A expression, and the E-cadherin results were unexpected. This finding could be attributed to the limited number of DDEC cases in this study. However, cases with p53 mutations in the UC had a poorer prognosis than those who did not. These findings suggest that MMR proteins and p53 may play a role in the dedifferentiation.

Whole-exome sequencing revealed that the WC and UC of DDEC exhibit different genetic mutations. Mutation signatures were also distinct between these components. This genetic heterogeneity is reminiscent of dedifferentiated liposarcoma [29], which also presents with mixed genetic profiles, contributing to a poor prognosis. In this study, new genetic mutations in the UC were uncommon across the three cases, making it challenging to identify a single gene responsible for dedifferentiation. Case 1 showed HRD, whereas cases 2 and 3 displayed MSI-H and hypermutator phenotypes, both of which are associated with an increased response to immune checkpoint inhibitors. Furthermore, approximately half of the UCs were MMR-deficient and expressed programmed death-ligand 1 (PD-L1), suggesting that DDEC could be a target for immune checkpoint inhibitors [30]. In case 2, mutations in *MSH6*, *ARID1A*, and *ARID1B* were observed in the UC, whereas case 3 exhibited mutations in *p53* and *POLE*. Common mutations in the UC of cases 2 and 3 suggest that *MSH6* and *ARID1A* may be involved in the dedifferentiation mechanism.

Fumarate hydratase (FH) is an enzyme involved in the tricarboxylic acid cycle and is a tumor suppressor gene [31]. *FH* mutations cause hereditary leiomyomatosis and renal cell carcinoma (HLRCC) [32]. In this study, case 2 exhibited a germline mutation in *FH*, which was also observed in the somatic mutation in case 3. However, *FH* mutations in both WCs and UCs suggest that *FH* does not play a role in the dedifferentiation process. Furthermore, p53 may be a driver of dedifferentiation [33]. The immunostaining results, particularly the presence of p53 mutations, supported the hypothesis that *TP53* dysfunction is involved in dedifferentiation.

Currently, platinum/taxane-based chemotherapy is recommended for DDEC [34]. However, in recent years, risk stratification based on molecular profiling and corresponding tailored management strategies have garnered increasing attention. In our present study, the three genetically analyzed cases exhibited diverse molecular subtypes according to the TCGA classification—copy number high, MSI-H, and POLE ultramutated. Although the short follow-up period may be a limiting factor, the copy number high case experienced recurrence and death, while the MSI-H and POLE ultramutated cases demonstrated favorable outcomes. These findings are consistent with the prognostic implications generally associated with each molecular subtype [35]. TCGA molecular classification of UDEC reveals that MMR deficiency occurs in 44.0%, POLE mutations in 12.4%, and p53 abnormalities in 18.6% [36]. The high prevalence of MSI and POLE groups suggests potential responsiveness to immunotherapy. A patient with recurrent DDEC was treated with pembrolizumab and gemcitabine and achieved 15 months of PFS [23]. Another patient with advanced-stage MMR-deficient DDEC was treated with three cycles of maintenance pembrolizumab and experienced over 5 years of PFS [37]. The results of the genetic analysis showed that HRD-related genes had numerous mutations. Poly (ADP-ribose) polymerase (PARP) inhibitors effectively treat HRD [38]. The DUO-E and DUO-O trials are currently underway, and the use of durvalumab and olaparib in maintenance therapy is expected to be effective [39,40]. For DDEC, the DUO-E trial regimen is considered the most appropriate because platinum/taxane-based chemotherapy effectively targets well-differentiated areas, immune checkpoint inhibitors are effective against MSI-H and hypermutator phenotypes, and PARP inhibitors are effective for HRD. APR-246 (Eprenetapopt) targets the p53 pathway [41]. In addition, the combination of adavosertib—a potent antitumor kinase inhibitor—and carboplatin has been used for advanced p53 mutated ovarian cancer in a phase II trial [42]. Targeted drugs for the *ARID1A* mutation exist, which is common in cases 2 and 3. Aurora A is a therapeutic target in ARID1A-deficient colorectal cancer cells [43]. Alisertib, an aurora A kinase inhibitor, has been investigated for the treatment of several cancers [44].

This study had several limitations. First, the study design was retrospective, and the number of DDEC cases was relatively small, which limits the statistical power to detect certain associations. For instance, some expected prognostic markers, such as E-cadherin or ARID1A loss, showed no significant effect on outcomes, likely due to the small sample size. Second, only three tumors underwent whole-exome sequencing (WES), which may have limited our ability to detect rare genomic drivers of dedifferentiation and precluded the identification of a single causative genetic event. Third, the results regarding natural incidence histology were based on data from a single institution, and multicenter studies are needed to validate and expand upon these findings.

## 4. Materials and Methods

### 4.1. Study Samples

A retrospective search of the pathology databases of the Seirei Hamamatsu General Hospital, Shimane University, and Shimane Prefectural Central Hospital was conducted between 2000 and 2020. Data from Shimane University from 2011 to 2020 were used to determine the natural incidence of histology. Tumors were histologically classified according to WHO criteria. A total of 273 cases of grade 1–2 endometrioid carcinoma, 41 grade 3 endometrioid carcinoma, 18 DDEC, 34 carcinosarcoma, and 49 other carcinomas (30 serous, 10 clear cell, 8 mucinous, and 1 squamous cell carcinoma) were identified. Five cases of carcinosarcoma and eleven of DDEC were collected from Seirei Hamamatsu General Hospital. Ten cases of carcinosarcoma and two of DDEC were collected from Shimane Prefectural Central Hospital. All the patients were initially treated surgically (modified radical hysterectomy + bilateral salpingo-oophorectomy ± pelvic lymphadenectomy ± para-aortic lymphadenectomy) and with adjuvant platinum-based chemotherapy (carboplatin; [AUC5], paclitaxel; 175 mg/m^2^, or docetaxel; 70 mg/m^2^). Systematic retroperitoneal lymphadenectomy was conducted in approximately 90% of patients. Postoperative adjuvant chemotherapy was undergone for patients with risk factors for recurrence (e.g., cervical invasion, deep myometrial invasion, lymphovascular space invasion, lymph node metastasis, and positive peritoneal cytology). A total of 415 patients were included in the survival analysis. The follow-up period ranged from 1 to 250 months (median, 60 months).

According to the International Federation of Gynecology and Obstetrics staging criteria (FIGO 2008 [45]), the number of DDEC cases in stages I, II, III, and IV were five (27.8%), 0 (0%), nine (50%), and four (22.2%), respectively. Patients’ information was retrospectively acquired from electronic medical records. Samples were collected after obtaining written informed consent from all patients, and the study was approved by the Shimane University Institutional Review Board (IRB No. 20070305-1 and No. 20070305-2, last update, 8 December 2019).

### 4.2. Immunohistochemistry

Immunohistochemistry was performed on 18 DDEC samples, and the expression of PAX8, cytokeratin, epithelial membrane antigen, p53, E-cadherin, fascin, ARID1A, and MMR proteins (MLH1, PMS2, MSH2, and MSH6) was evaluated. These markers were selected based on previous studies indicating their prognostic relevance in DDEC [10,11]. Detailed antibody information is provided in Table 4. Staining procedures followed the manufacturer’s instructions [46,47,48], and evaluations were conducted by two researchers (H.H. and M.N.) using a double-blind method.

We defined immunohistochemical staining results as follows: the complete absence of staining was considered negative, whereas staining observed in 1% or more of tumor cells was considered positive. For p53, diffuse and strong nuclear staining was interpreted as overexpression (mutant), while all other staining patterns were classified as wild-type expression.

### 4.3. DNA Extraction

Next-generation sequencing was performed on three DDEC samples [49,50]. For total DNA isolation, DNA was extracted from formalin-fixed, paraffin-embedded tissues as previously mentioned [50]. The WC and UC were separately collected using macroscopic hematoxylin and eosin staining, and normal tissue was used as a control.

### 4.4. Statistical Analyses

Statistical analyses were conducted using SPSS software (version 19.0; IBM Corporation, Armonk, NY, USA). Kaplan–Meier curves and log-rank tests were used to compare the progression-free survival (PFS) and overall survival (OS). PFS was calculated from the first day of treatment to the date of recurrence or last follow-up, whereas OS was defined from the first day of treatment to the date of death or last follow-up. Chi-square tests were used to compare clinicopathological characteristics across different histological types. Paired sample *t*-tests were used to compare the immunostaining results of the WC and UC. Statistical significance was set at *p* < 0.05.

## 5. Conclusions

This study presents the first accumulation of data on DDEC in the Japanese population and provides valuable insights into its characteristics. Genetic analysis revealed that approximately one-third of the cases exhibited HRD, whereas two-thirds showed MSI-H and hypermutator phenotypes, suggesting that immune checkpoint and PARP inhibitors may be effective treatments for DDEC. Additionally, the high prevalence of p53 mutations detected by immunohistochemistry supports the use of drugs that target the p53 pathway. However, larger surveys are needed in the future to verify our hypothesis. Future studies will create an organoid and xenograft [51] of DDEC to use as models for chemosensitivity tests and genotype-matched therapy. Furthermore, to enable the personalized management of each DDEC case according to its molecular diversity, an ongoing project is investigating the TCGA molecular classification of DDEC in the Japanese population.

## Figures and Tables

**Figure 1 ijms-26-04137-f001:**
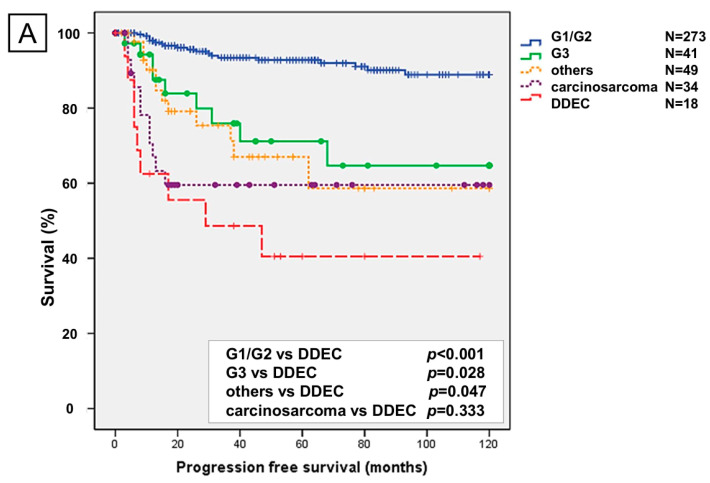
Kaplan–Meier analysis of progression-free survival (**A**) and overall survival (**B**). G1/G2, grade 1 and 2 endometrioid carcinoma; G3, grade 3 endometrioid carcinoma; others, other carcinomas; DDEC, dedifferentiated endometrioid carcinoma.

**Figure 2 ijms-26-04137-f002:**
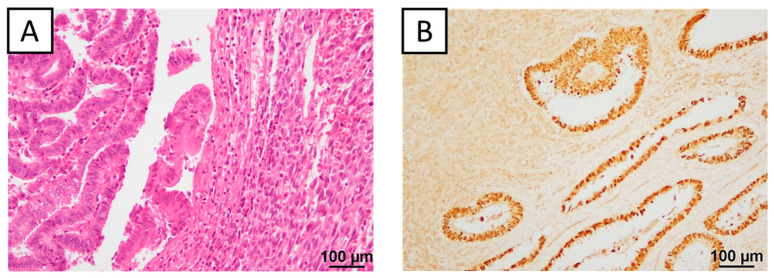
Immunohistochemical findings. (**A**) Hematoxylin and eosin staining showing WCs and UCs from case 9. (**B**) PAX8: expression in the WC and the loss of expression in the UC in case 7. (**C**) Cytokeratin: expression in the WC and the loss of expression in the UC in case 9. (**D**) Epithelial membrane antigen: expression in the WC and the loss of expression in the UC in case 9. (**E1**) p53: wild-type expression in the WC and UC in case 9. (**E2**) p53: overexpression in the WC and UC in case 7. (**F**) E-cadherin: expression in the WC and the loss of expression in the UC in case 7. (**G**) Fascin: the loss of expression in the WC and expression in the UC in case 9. (**H**) ARID1A: expression in the WC and the loss of expression in the UC in case 9. WC, well-differentiated component; UC, undifferentiated component.

**Figure 3 ijms-26-04137-f003:**
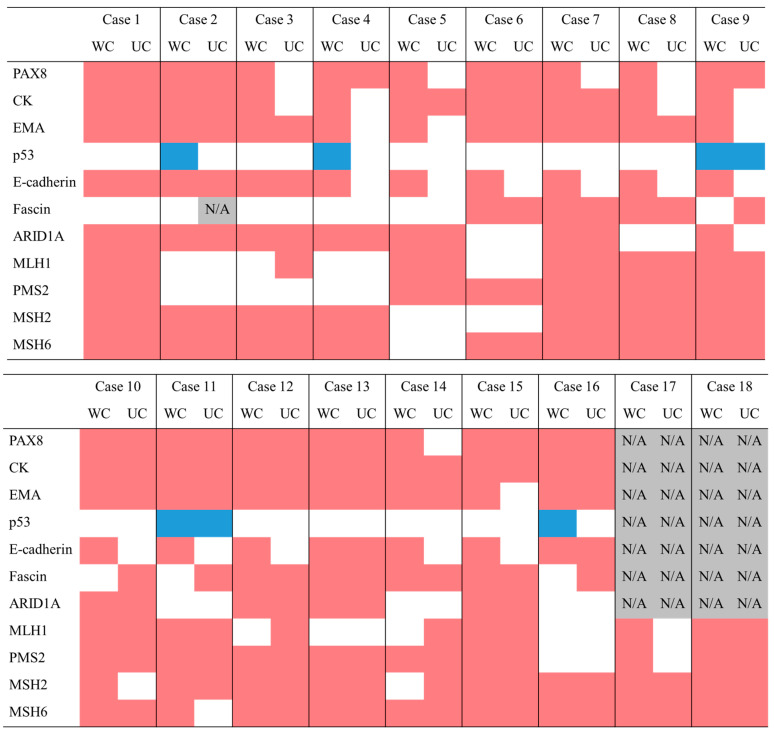
Summary of immunohistochemical findings. WC, well-differentiated component; UC, undifferentiated component; CK, cytokeratin; EMA, epithelial membrane antigen; N/A, not performed. Positive results are indicated in red, and negative results are indicated in white (for p53, wild-type is blue, mutant is white).

**Figure 4 ijms-26-04137-f004:**
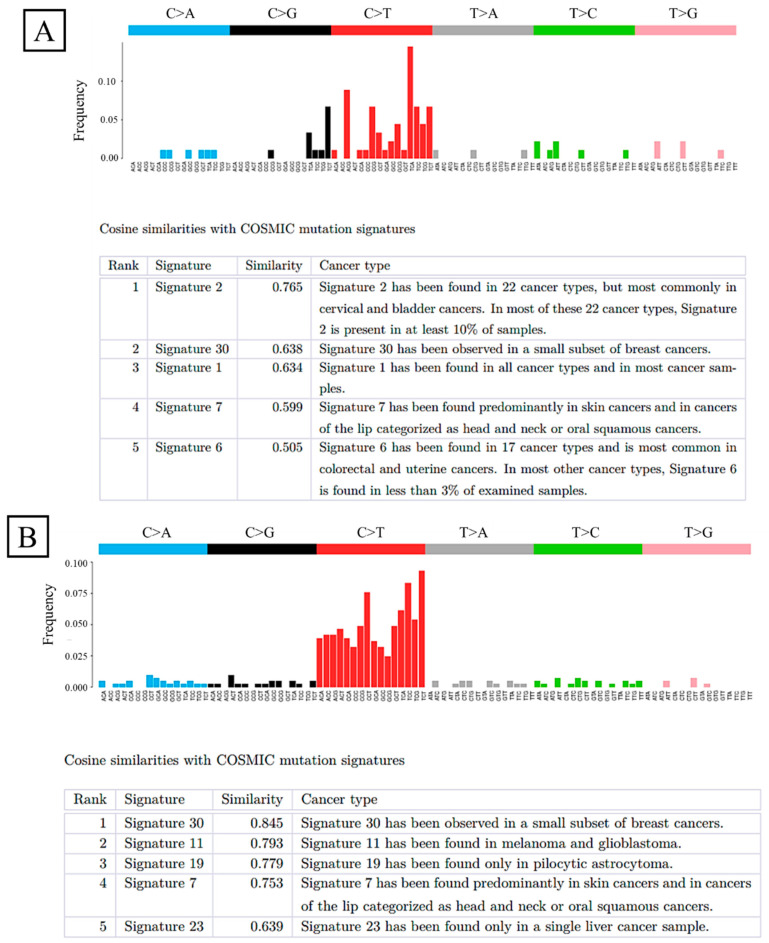
Mutation signatures. (**A**) Signature 2 exhibits the highest similarity with the mutation profile of the WC in case 1. (**B**) Signature 30 exhibits the highest similarity with the mutation profile of the UC in case 1. WC, well-differentiated component; UC, undifferentiated component.

**Figure 5 ijms-26-04137-f005:**
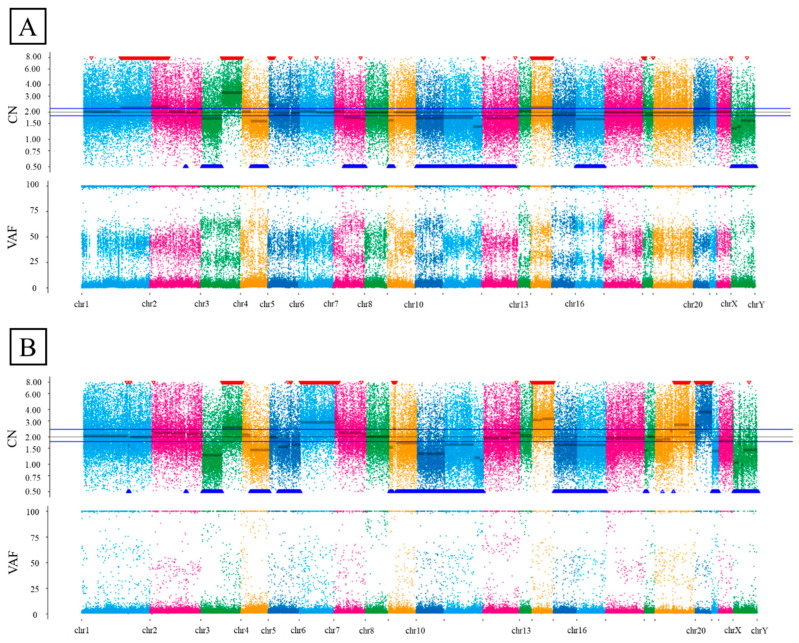
The copy number alteration plot. The horizontal axis represents the chromosome location, and the vertical axis represents the gene copy number. (**A**) The CNA plot of the WC in case 1. (**B**) The CNA plot of the UC in case 1. CN, copy number; CNA, copy number alteration; VAF, variant allele frequency; WC, well-differentiated component; UC, undifferentiated component.

**Table 1 ijms-26-04137-t001:** Natural incidence histology of endometrial cancer.

Histology	Number of Patients (n = 255)	Percentage (%)
Endometrioid carcinoma G1/G2	188	73.7
Endometrioid carcinoma G3	23	9.0
Others	39	15.3
Serous carcinoma	28	11.0
Clear cell carcinoma	6	2.4
Mucinous carcinoma	5	2.0
Dedifferentiated endometrioid carcinoma	5	2.0

**Table 2 ijms-26-04137-t002:** Clinicopathological characteristics of each histological type.

Characteristic	G1/G2	*p*-Value	G3	*p*-Value	Others	*p*-Value	Carcino Sarcoma	*p*-Value	DDEC
Age (%)		0.554		0.931		0.102		0.053	
<60	156 (57)		20 (49)		14 (29)		8 (24)		9 (50)
≥60	117 (43)		21 (51)		35 (71)		26 (76)		9 (50)
FIGO stage (%)		<0.01		0.011		0.024		0.236	
I, II	236 (87)		27 (66)		30 (61)		16 (48)		5 (28)
III, IV	36 (13)		14 (34)		19 (39)		17 (52)		13 (72)
Muscle invasion (%)		<0.01		0.032		<0.01		0.055	
<50%	195 (72)		17 (42.5)		28 (61)		6 (46)		2 (12.5)
≥50%	77 (28)		23 (57.5)		18 (39)		7 (54)		14 (87.5)
LVSI (%)		<0.01		0.244		0.036		0.317	
Yes	105 (39)		29 (74)		27 (59)		12 (100)		14 (87.5)
No	162 (61)		10 (26)		19 (41)		0 (0)		2 (12.5)
Pelvic/para aortic lymph metastasis (%)		<0.01		0.297		0.213		0.170	
Yes	26 (10)		12 (29)		13 (27)		9 (69)		7 (44)
No	247 (90)		29 (71)		35 (73)		4 (31)		9 (56)

G1/G2, endometrioid carcinoma grade 1 and grade 2; G3, endometrioid carcinoma grade 3; Others, other carcinomas; DDEC, dedifferentiated endometrioid carcinoma; LVSI, lymphovascular space invasion.

**Table 3 ijms-26-04137-t003:** Integrated molecular and clinical profiles of three DDEC cases.

		LOH (%)	TMB	MSI (%)	TCGA	PFS (Months)	OS (Months)
Case1	WC	21.57	-	-	Copy number high	17 recurrence	27 death
	UC	22.371	-	-			
Case2	WC	-	227	30.25	MSI-H	38 *	38 *
	UC	-	1927	36.38			
Case3	WC	-	1099	-	POLE ultramutated	11 *	11 *
	UC	-	15,105	28.88			

DDEC, dedifferentiated endometrioid carcinoma; WC, well-differentiated component; UC, undifferentiated component; LOH, loss of heterozygosity; TMB, tumor mutation burden; MSI-H, microsatellite instability-high; TCGA, The Cancer Genome Atlas; PFS, progression-free survival; OS, overall survival. The unit of tumor mutation burden is nonsynonymous mutation. * Censored at the time of last follow-up.

**Table 4 ijms-26-04137-t004:** Details of immunohistochemical procedure.

Antibody	Producer	Dilution
PAX8	Proteintech (Chicago, IL, USA) (10336-1-AP)	1:500
CK	Santa Cruz Biotechnology (Santa Cruz, CA, USA) (sc-8018)	1:50
EMA	Thermo Fisher Scientific (Waltham, MA, USA) (MA1-06503)	1:100
p53	Dako (Santa Clara, CA, USA) (M7001)	1:50
E-cadherin	Abcam (Cambridge, UK) (ab15148)	1:50
Fascin	Thermo Fisher Scientific (MAF-11483)	1:200
ARID1A	Santa Cruz Biotechnology (sc-32761)	1:100
MLH1	Dako (M3640)	1:50
PMS2	Dako (M3647)	1:40
MSH2	Dako (M3639)	1:50
MSH6	Dako (M3646)	1:50

CK, cytokeratin; EMA, epithelial membrane antigen.

## Data Availability

The data that support the findings of this study are available from the corresponding authors (K.N.) (S.K.) upon reasonable request.

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
