# Peer review of "Exploring the Genetic and Clinical Landscape of Dedifferentiated Endometrioid Carcinoma"

_ijms, 2025, doi:10.3390/ijms26094137_

Round 1
Reviewer 1 Report
Comments and Suggestions for Authors
Haraga et al. analysed survival rates of various histotypes of endometrial cancers, collected FFPE samples from 18 DDEC cases, and compared the UC component with the WC one in each sample in terms of expression of different marker proteins as revealed by immunohistochemistry. Furthermore, they performed WES for UC and WC parts of each of 3 DDEC cases. Although the number of cases is small, the authors’ point of view is interesting and their interpretation of the results seems reasonable. However, looking at Figure 3, I am not sure whether several DDECs judged by the authors are true. The UC component of typical DDEC should express neither PAX8 nor CK but EMA to some extent. According to WHO’s definition of DDEC, it seems that DDEC cases are only Cases 3 and 8 in Figure 3. At least, Cases 1, 2, 4, 6, 7, 10, 11, 12, 13, 14, 15, and 16 look like G3 but not DDEC. The authors should clarify definition of “positive” and “negative” in Figure 3. Otherwise, I do not know what they are discussing about in the paper.
Author Response
Point-wise responses to the comments made by Reviewer 1:
Comments and Suggestions for Authors
Q1)
Haraga et al. analysed survival rates of various histotypes of endometrial cancers, collected FFPE samples from 18 DDEC cases, and compared the UC component with the WC one in each sample in terms of expression of different marker proteins as revealed by immunohistochemistry. Furthermore, they performed WES for UC and WC parts of each of 3 DDEC cases. Although the number of cases is small, the authors’ point of view is interesting and their interpretation of the results seems reasonable.
A1)
Thank you for your kind consideration.
Q2)
However, looking at Figure 3, I am not sure whether several DDECs judged by the authors are true. The UC component of typical DDEC should express neither PAX8 nor CK but EMA to some extent. According to WHO’s definition of DDEC, it seems that DDEC cases are only Cases 3 and 8 in Figure 3. At least, Cases 1, 2, 4, 6, 7, 10, 11, 12, 13, 14, 15, and 16 look like G3 but not DDEC. The authors should clarify definition of “positive” and “negative” in Figure 3. Otherwise, I do not know what they are discussing about in the paper.
A2)
We defined “negative” as complete absence of staining in tumor cells, and “positive” as any staining observed in 1% or more of tumor cells. For p53, diffuse and strong nuclear staining was interpreted as overexpression (mutant), whereas all other staining patterns were considered as wild-type. These definitions have now been clearly described in the “Immunohistochemistry” section of the Materials and Methods.
According to the 2020 WHO Classification, immunohistochemistry is regarded as a desirable tool in the diagnosis of dedifferentiated endometrioid carcinoma (DDEC). However, not all cases exhibit typical immunophenotypic features, and there remain aspects of DDEC pathology that are still not fully understood. In this context, we aimed to explore and characterize the diversity of immunohistochemical expression patterns in our DDEC cohort through this study.
Reviewer 2 Report
Comments and Suggestions for Authors
The authors provide a significant contribution by integrating thorough molecular analysis with clinical insights for a rare but highly aggressive form of endometrial cancer. What needs be highlighted as a novelty -in addition to the reporting of the first DDEC Japanese cohort is the DDEC diversity; it is not a uniform entity and the linking of these rare tumours to established molecular subtypes of endometrial cancer. On that note, they should speculate a bit more on how their cases would be managed differently? in the current era of risk stratification by molecular profiling. I appreciate the NCNN guidelines on DDEC but the study needs be fully align with contemporaneous considerations as there are studies out there aiming to classify DDEC by molecular subgrouping (PMID: 31811476). In that case, I wonder whether the title of the manuscript should be more concise to highlight the novelty.
What is also striking is the identification of an ultramutated POLE-mutant case: a POLE-mutation usually predicts favorable prognosis, but in this study a POLE-mutant tumor behaved aggressively, suggesting that dedifferentiation can override current molecular prognostic markers – what an insight to refine our understanding of endometrial cancer behaviour.
Balanced against all its strengths, the study has few limitations. The cohort size is modest, limiting the power to detect certain associations. For instance, some expected prognostic markers (such as E-cadherin or ARID1A loss) showed no significant effect on outcomes, likely due to the small sample​. Only three tumors underwent WES, which means that rare genomic drivers of dedifferentiation could have been missed. This could not allow for a single causative genetic event to be definitively pinpointed. Please comment.
Author Response
Point-wise responses to the comments made by Reviewer 3:
Comments and Suggestions for Authors
Q1)
The authors provide a significant contribution by integrating thorough molecular analysis with clinical insights for a rare but highly aggressive form of endometrial cancer.
A1)
Thank you for your kind consideration.
Q2)
What needs be highlighted as a novelty -in addition to the reporting of the first DDEC Japanese cohort is the DDEC diversity; it is not a uniform entity and the linking of these rare tumours to established molecular subtypes of endometrial cancer. On that note, they should speculate a bit more on how their cases would be managed differently? in the current era of risk stratification by molecular profiling. I appreciate the NCNN guidelines on DDEC but the study needs be fully align with contemporaneous considerations as there are studies out there aiming to classify DDEC by molecular subgrouping (PMID: 31811476).
A2)
Considering the diversity of DDEC, individualized management is essential, and molecular profiling is believed to play a crucial role. As part of the risk stratification based on molecular profiling, we placed emphasis on the TCGA molecular classification. We added the TCGA subtype and corresponding prognosis for each case to Table 3, and included a discussion of these findings in the revised manuscript. To support the prognostic relevance of TCGA subtypes, we included Reference 40. We believe that investigating the TCGA molecular classification in the Japanese DDEC population is of great interest, and we have mentioned that this is currently an ongoing project.
Q3)
In that case, I wonder whether the title of the manuscript should be more concise to highlight the novelty.
A3)
Thank you for your valuable suggestion. In line with your comment, we considered using a more concise title such as "Molecular Diversity of Dedifferentiated Endometrioid Carcinoma" to highlight the novelty. However, since our study places emphasis not only on genetic analysis but also on clinical characteristics, we have decided to retain the original title.
Q4)
What is also striking is the identification of an ultramutated POLE-mutant case: a POLE-mutation usually predicts favorable prognosis, but in this study a POLE-mutant tumor behaved aggressively, suggesting that dedifferentiation can override current molecular prognostic markers – what an insight to refine our understanding of endometrial cancer behaviour.
A4)
Our POLE-mutated case demonstrated a favorable outcome, with no recurrence or death observed during the follow-up period, although the duration of follow-up was relatively short. However, we acknowledge that recurrence can occur in clinical practice, and we are aware that genomic medicine is being increasingly applied in such cases.
Q5)
Balanced against all its strengths, the study has few limitations. The cohort size is modest, limiting the power to detect certain associations. For instance, some expected prognostic markers (such as E-cadherin or ARID1A loss) showed no significant effect on outcomes, likely due to the small sample​. Only three tumors underwent WES, which means that rare genomic drivers of dedifferentiation could have been missed. This could not allow for a single causative genetic event to be definitively pinpointed. Please comment.
A5)
Thank you for your valuable comment. As suggested, we have revised the Limitations section to include the limitations regarding the modest cohort size, lack of significant findings for expected prognostic markers (such as E-cadherin or ARID1A loss), and the limited number of tumors subjected to whole-exome sequencing (WES). We have also mentioned the potential impact of these factors on the ability to detect rare genomic drivers and to pinpoint a single causative genetic event.
Reviewer 3 Report
Comments and Suggestions for Authors
this study aimed to characterize DDEC by collecting data from Japanese patients and identifying
genetic abnormalities involved in the dedifferentiation process. The study is interesting and provide useful and novel information with only one limitation of the low cases number, but this out of their hand due to the rare incidence. Some minor comments, if possible, should be made before final acceptance
Line 111: add “respectively” after the last p value.
Line 112: if the p value is more than 0.05 to 0.09; it is better to be described to “tend to be higher” not directly higher. Please apply this on your p value (p =0.083).
- I think table 1 doesn’t add more information than the text. Please use one of them, either the text or the table.
Author Response
Point-wise responses to the comments made by Reviewer 3:
Comments and Suggestions for Authors
Q1)
This study aimed to characterize DDEC by collecting data from Japanese patients and identifying genetic abnormalities involved in the dedifferentiation process. The study is interesting and provide useful and novel information with only one limitation of the low cases number, but this out of their hand due to the rare incidence. Some minor comments, if possible, should be made before final acceptance.
A1)
Thank you for your kind consideration.
Q2)
Line 111: add “respectively” after the last p value.
A2)
Thank you for your comment. We have added the word “respectively” after the last p-value in line 111, as suggested.
Q3)
Line 112: if the p value is more than 0.05 to 0.09; it is better to be described to “tend to be higher” not directly higher. Please apply this on your p value (p =0.083).
A3)
Thank you for your valuable suggestion. In accordance with your comment, we have revised the description to indicate that the proportion of p53 mutations tended to be higher in UC (87.5%) than in WC (68.6%) (p = 0.083).
Q4)
I think table 1 doesn’t add more information than the text. Please use one of them, either the text or the table.
A4)
Thank you for your helpful comment. Following your suggestion, we have removed the text and retained Table 1 to avoid redundancy and improve clarity.
Round 2
Reviewer 1 Report
Comments and Suggestions for Authors
Taking into account my previous comments and requests, the authors revised their manuscript appropriately.
I recommend the manuscript be published.